



# Spatial analysis of PM₂.₅ using a Concentration Similarity Index applied to air quality sensor networks

Rósín Byrne[1, 2], John. C. Wenger[1, 2], Stig Hellebust[1, 2].

[1]Centre for Research into Atmospheric Chemistry, School of Chemistry, University College Cork, Ireland.
5  [2]Environmental Research Institute, University College Cork, Ireland.

*Correspondence to*: Stig Hellebust (s.hellebust@ucc.ie)

**Abstract.** Air quality sensor (AQS) networks are useful for mapping $PM_{2.5}$ in urban environments, but quantitative assessment of the observed spatial and temporal variation is currently under-developed. This study introduces a new metric - the Concentration Similarity Index (CSI) - to facilitate a quantitative and time-averaged comparison of the 10  concentration-time profiles of $PM_{2.5}$ measured by each sensor within an air quality sensor network. Following development on a dataset with minimal unexplained variation and robust tests, the CSI function is ensured to represent an unbiased and fair depiction of the air quality variation within an area covered by a monitoring network. The measurement data is used to derive a CSI value for every combination of sensor pairs in the network, which can then be compared with others in the network, yielding valuable information on spatial variation in $PM_{2.5}$. This new method is applied to two separate AQS 15  networks in Dungarvan and Cork City, Ireland. Dungarvan yielded a lower mean CSI, indicating lower overall similarity between locations in the network, possibly due to the town's coastal location giving rise to higher variation within the network. In both networks, the average diurnal plots for each sensor exhibit an evening peak in $PM_{2.5}$ concentration due to emissions from residential solid fuel burning, however, there is considerable variation in the size of this peak. Clustering techniques applied to the CSI matrices identify two different location types in each network; locations in central or 20  residential areas which experience more pollution from sold fuel burning and locations on the edge of the urban areas which experience cleaner air. Furthermore, the examination of isolated data periods (January and May) indicates higher $PM_{2.5}$ levels during periods of increased residential solid fuel burning act as a major driver for greater differences (lower similarity indices) between locations in both networks. Additionally, the CSI method facilitates the assessment of the representativeness of the $PM_{2.5}$ measured at regulatory air quality monitoring locations with respect to population exposure, 25  showing here that location type is more important than physical proximity in terms of similarity assessment. Applying the CSI in this manner can allow for the placement of monitoring infrastructure to be optimised. The findings of this work underscore the influence of solid fuel combustion as a local contributor to $PM_{2.5}$ and the variation it can cause between the measurements at different monitoring locations in a network while also highlighting the importance of including wintertime PM data for accurate comparisons. The CSI method developed here could be a valuable tool for quantitative comparisons of 30  air quality within a monitoring network, offering insights for further regulatory monitoring and exposure assessments.



## 1 Introduction

Air pollution affects the environment, quality of life and is a major cause of premature death and disease (Lelieveld et al., 2015; Cesaroni et al., 2013; Raaschou-Nielsen et al., 2013; Pedersen et al., 2013). The category of air pollutant with the largest impact on human mortality and health is fine particulate matter, i.e. atmospheric particles with an aerodynamic diameter of 2.5 micrometres or less ($PM_{2.5}$) (Pope et al., 2020; Samoli et al., 2013; Pope and Dockery, 2012). In many regions around the world, air quality monitoring and management have become critical endeavours to mitigate the detrimental effects of air pollution, and especially $PM_{2.5}$, on citizens and the environment.

Over the years, technological advances have provided valuable tools to enhance our understanding of air pollution, and low-cost air quality sensors (AQS) are emerging as promising instruments for collecting real-time air quality data at an improved spatial and temporal resolutions (Kumar et al., 2015; Munir et al., 2019). When used in networks, air quality sensors offer immense potential for enhancing and supplementing regulatory monitoring and assessment (Malings et al., 2020). However, further work needs to be carried out to assess the effectiveness of sensor networks and how to make best use of the data for gaining further insights into air pollution within a locality. Careful consideration must be given to the quality of the data provided by sensors and the requirement for calibration must be assessed (Diez et al., 2022). Recent studies have shown that the performance and calibration of a $PM_{2.5}$ sensor is dependent on the type of sensor and often on the measurement location, suggesting the need for site-specific and individual calibrations (Wang et al., 2015; Zamora et al., 2020; Kaur and Kelly, 2023; Sayahi et al., 2019). When these factors are considered and accounted for, AQS networks offer an unprecedented opportunity to gain further insights into the complex dynamics of air pollution in localised areas, such as urban environments, industrial zones, and residential neighbourhoods (Hodoli et al., 2023; Heimann et al., 2015; Crawford et al., 2021; O'Regan et al., 2022; Frederickson et al., 2022).

Assessing the spatial variation of air quality is of paramount importance because air pollution is not homogenous and can exhibit significant variations across different areas even on a local scale (Kassomenos et al., 2014; Wang et al., 2018; Frederickson et al., 2023, 2022). The variability of air pollution can be influenced by a multitude of factors such as traffic patterns, industrial activities, meteorological conditions, and local topography. Consequently, relying on single monitoring locations or limited data resolution can provide an incomplete picture and inadequate understanding of local air quality in a certain area (Li et al., 2019). Understanding these variations is crucial for targeted interventions and policy decisions aimed at improving air quality and safeguarding public health. Spatial analysis, facilitated by sensor networks allows for a more accurate and nuanced understanding of how air quality, and therefore exposure to pollution, varies across a population centre.

In a recent study, we used data collected by a $PM_{2.5}$ sensor network in the city of Cork, Ireland, to estimate the contribution of local pollution sources as separate and distinct from regional or transported air pollution (Byrne et al., 2023). The results highlighted the very localised nature of $PM_{2.5}$ caused by residential solid fuel burning during winter, which is a significant





problem in many towns and cities in Ireland and elsewhere (Dall'osto et al., 2013; Ovadnevaite et al., 2021; Wenger et al., 2020; Kourtchev et al., 2011; Lin et al., 2018; Zhang et al., 2021; Lin et al., 2019).

In this work, we propose a new approach for assessing the spatial profile of air quality using an AQS network. The method yields a time-averaged concentration similarity index (CSI) for quantitative assessment of the similarity between the complete data series produced by different sensors within the network. The CSI is built on the premise that sensors exposed to similar ambient conditions and pollutant sources will produce comparable $PM_{2.5}$ temporal trends. Conversely, sensors subject to different conditions might display divergent $PM_{2.5}$ concentration trends. The motivation for the development of an assessment method based on the temporal variation over an extended period is the realisation that the annual average is a poor representation of true population exposure, which is experienced from hour to hour and day to day. It is therefore not adequate to merely compare annual averages of $PM_{2.5}$ levels in different locations in order to compare the exposures to $PM_{2.5}$ experienced by the populations in the respective locations. This method aims to translate this idea into a quantifiable metric by calculating the time-averaged degree of similarity between two sensor datasets. Method development and testing is performed on an AQS network in the town of Dungarvan in Ireland to identify areas that may be experiencing persistently elevated or very localised $PM_{2.5}$ pollution compared to others. Clustering techniques are used to group sensors based on the similarity of their $PM_{2.5}$ measurements. The CSI method is also retrospectively applied to the data collected in the Cork City network to investigate the transferability of the method between sensor networks and to explore any differences between the locations.

## 2 Methodology

### 2.1 Data collection, preprocessing, and calibration

The collection, preprocessing, and calibration of the data collected by the $PM_{2.5}$ sensor networks in Dungarvan and Cork City was carried out using the Julia programming language (Bezanson et al., 2017). Since low-cost AQS are not of regulatory standard, great care needs to be taken with quality assessment and quality control of the data. In particular, the degree to which changes or differences in $PM_{2.5}$ measurements between devices can be trusted needs to be considered. The methodology proposed here addresses these inherent issues to deliver an approach for assessing the spatial representativeness of any monitoring location, i.e. what is the extent of the geographical area that the location meaningfully represents, in air quality monitoring terms, and what environment type is it comparable to regardless of geographical distance to the location?

### 2.1.1 Dungarvan $PM_{2.5}$ sensor network

The Dungarvan sensor network consisted of 18 solar powered Clarity Node-S devices (Clarity Movement Co., USA) which utilise the Plantower PM6003 sensor to measure $PM_{2.5}$ within the range 1-1000 $\mu g\ m^{-3}$ and at a resolution of 1 $\mu g\ m^{-3}$ (Clarity Movement Co., 2023; Node-S technical sheet, 2023). By default, the Node-S devices take measurements every



15 minutes, allowing sufficient data upload and battery sleep time in between sampling periods. However, this can be adjusted to higher or lower frequencies. The highest sampling frequency achievable during winter without significantly affecting the battery performance was 8 minutes.

The Clarity Node-S devices were typically attached to street light poles between 2 and 4 metres above the ground. The sensors were positioned in a range of different environments including urban background, residential, coastal, and roadside

locations (Figure S1). Many of these locations were a mix of the different environments. The majority of devices were operational from 1 November 2022 to 31 May 2023, however three devices (AP7, AY9N, AY93) were only deployed from 12 January 2023. Measurements were taken over a continuous period covering different meteorological seasons (mainly Winter and Spring/early Summer), thus ensuring temporal variations in $PM_{2.5}$ concentrations were captured comprehensively.

Prior to and after deployment in Dungarvan, the Clarity Node-S devices were co-located on the roof of the Ellen Hutchins Building, University College Cork (51.895136, -8.516146) to compare their performance. Details of the three co-location periods are outlined in Table S1. Although some devices were not available for all three co-location periods, the three periods combined provide a comparison between the sensors across different seasons. This co-location dataset enabled the CSI method to be developed on measurements that in theory should be equal and the function could then be modified, if

necessary, to allow for sensor behaviour, uncertainties, errors, and potential limitations.

The raw sensor data from the co-location periods and field deployment, underwent a series of preprocessing steps to mitigate potential sources of error in the measurement and ensure data quality and consistency. Data points outside of the operational range of the sensors (> 1000 µg m$^{-3}$) were identified and removed, although instances of these were minimal. The 8-minute data were averaged to produce hourly measurements. Missing data points could potentially affect the temporal continuity of

the data; however, the data coverage was overall very good for the co-location and measurement campaign periods. On average, the devices had an hourly measurement coverage of 87 % for the field measurement campaign. This corresponds to an average of 4443 hourly measurements per device for the campaign period.

Assessing the consistency of measurements across the sensor network was paramount. Although the $PM_{2.5}$ readings were very well correlated when the devices were co-located (Table S2), a data harmonisation procedure was performed to ensure

the uniformity of sensor measurements, which is a prerequisite for the subsequent development of the Concentration Similarity Index. Since there was no reference-grade $PM_{2.5}$ data available during the co-location periods, the $PM_{2.5}$ concentrations from each sensor were scaled to a common reference point, represented by the mean of all data points across the whole co-location dataset (Figure S2). The data series for each sensor was then individually compared with the calculated mean dataset and subsequently harmonised to the common reference point using a simple linear regression

approach. The equations resulting from this harmonisation procedure were applied to the measurements collected from all devices during the subsequent field measurement campaign. While this procedure did not convert the measured $PM_{2.5}$ to reference-equivalent concentrations, it minimised sensor output variability and facilitated a more equitable comparison between sensor measurements (Table S2).



### 2.1.2 Cork City PM₂.₅ sensor network

The Cork City sensor network consisted of 16 PurpleAir PA-II-SD units which each contain two Plantower PMS5003 sensors to measure $PM_{2.5}$ within the effective range 0-500 μg m⁻³, with a maximum range of 1000 μg m⁻³, and at a resolution of 1 μg m⁻³ (PMS5003 series data manual, 2022). In this study, data recorded by the devices in the network for the periods 01 January 2021 to 31 May 2021 and 1 September 2021 to 31 December 2021 were collated and analysed. However, four devices were found to have limited data capture for the specified periods (< 50 %) and were therefore omitted from the

analysis. The 12 sensors used in this analysis had an average data capture of 85 % for the specified periods; their locations are shown in Fig. S3.

Due to logistical constraints, it was not possible to co-locate all of the PurpleAir devices together to assess variability in $PM_{2.5}$ concentrations. However, low inter-sensor and inter-unit variability was exhibited by four co-located PurpleAir devices in our previous study on the Cork City network (Byrne et al., 2023). Moreover, the $PM_{2.5}$ concentrations measured

using the four PurpleAir devices were highly correlated with hourly values of $PM_{2.5}$ obtained using a Met-One (USA) Beta-Attenuation Monitor (BAM-1020) and a co-location dataset was used to derive calibration factors incorporating the effects of temperature and relative humidity. The data processing procedures for obtaining the $PM_{2.5}$ concentrations reported here are identical to those reported by Byrne et al. (2023).

The Cork City dataset spans a similar measurement period to the Dungarvan dataset to allow for comparable results due to

the known seasonality of $PM_{2.5}$ pollution in Ireland (Ovadnevaite et al., 2021). Although the year 2021 included some periods of COVID-19 pandemic restrictions, such measures mainly affected $NO_2$ concentrations and were not shown to have a significant impact on PM levels in Ireland (Environmental Protection Agency (EPA), 2020).

### 2.2 Development of the Concentration Similarity Index

The Concentration Similarity Index (CSI) derived here quantifies the degree of likeness between $PM_{2.5}$ concentration profiles

from two sensors for a defined period of time and forms the basis for assessing the spatial disparities in $PM_{2.5}$ measurements within sensor networks. The methodology proposed was developed through multiple iterations in order to adjust and improve the procedure. An overview of the development is described, showing the evolution towards the final method.

### 2.2.1 Original function application

The first phase of development was based directly on the work carried out by Piersanti et al. (2015), who used a

concentration similarity function to assess the spatial representativeness of $PM_{2.5}$ and $O_3$ monitoring stations in the Italian air quality monitoring network. By comparing point measurements to a dataset of modelled hourly air pollutant data covering Italy with a $4 \times 4$ km² grid cell resolution, Piersanti et al. (2015) produced maps showing how representative certain sites in the Italian monitoring infrastructure were. The application proposed here compares point measurement to point measurement as opposed to point measurement to modelled grid cell data, however the underlying principle of comparing two





concentration-time profiles to produce a single indication of similarity between them still applies. The function value $f_{site}(x, y)$ used by Piersanti et al. (2015) to assess the spatial coverage of point measurements is given in Eq. (1):

$$f_{site}(x,y) = \frac{\sum_{i=1}^{N_t} flag}{N_t}, where\ flag = \begin{cases} 1, & \frac{|C(X_{site}, Y_{site}, t_i) - C(x, y, t_i)|}{C(X_{site}, Y_{site}, t_i)} < 0.2 \\ 0, & \frac{|C(X_{site}, Y_{site}, t_i) - C(x, y, t_i)|}{C(X_{site}, Y_{site}, t_i)} > 0.2 \end{cases}$$

$$(1)$$

Where, $C(x, y, t_i)$ represents the surface concentration from the modelled data at time $t_i$, $C(X_{site}, Y_{site}, t_i)$ represents the point measurement of a specific monitoring site at time $t_i$, and $N_t$ is the total number of time steps. The study defined a point measurement as representative of a grid cell area if the condition $f_{site}(x, y) > 0.9$ is true.

In the first step of our approach, this function was applied to the hourly average PM$_{2.5}$ data obtained from the co-located Clarity Node-S units by comparing two sensor data series at a time, with the reference and modelled concentration inputs

substituted for sensor PM concentration values, $C(A, t_i)$ and $C(B, t_i)$. Over a total of 1565 co-located hours, the mean number of comparable data points per $C(A, t_i)$, $C(B, t_i)$ pair was 654, due to devices being present at different stages during the co-location periods (Table S1).

It might be expected that the function value comparing two sensor data series would be 1, given that the measurements were collected in the same location and were known to all represent the same air parcel at each point in time. However, it was

found that the function was not comprehensive enough to allow for an acceptable comparison of the sensor data. While in theory, the results for all sensor pairs should be 1, the results showed discrepancies between some device pairs, because the function value deviated significantly from 1 in many cases (Table 1) and was as low as 0.51 in some cases, with an overall mean of 0.82.






**Table 1: Function values, $f_{site}(x, y)$, for hourly averaged PM$_{2.5}$ measured by a range of co-located Clarity Node-S devices. Device labels in the columns were set as $C(X_{site}, Y_{site}, t)$ and device labels in the rows were set at $C(x, y, t_i)$.**

| | A3 | A4 | A8H | A8Z | A9 | AQ | AZ | A7 | A6P | AJ3 | AP7 | AQV | ARF | AW6 | AWF | AY9N | AY93 | AYG |
|---|---|---|---|---|---|---|---|---|---|---|---|---|---|---|---|---|---|---|
| **A3** | 1 | 0.83 | 0.88 | 0.8 | 0.8 | 0.86 | 0.88 | 0.87 | 0.87 | 0.99 | 0.67 | 0.9 | 0.8 | 0.99 | 0.8 | 0.61 | 0.8 | 0.98 |
| **A4** | 0.84 | 1 | 0.81 | 0.89 | 0.83 | 0.87 | 0.89 | 0.87 | 0.87 | 0.98 | 0.8 | 0.9 | 0.9 | 0.97 | 0.85 | 0.72 | 0.79 | 0.99 |
| **A8H** | 0.86 | 0.79 | 1 | 0.79 | 0.78 | 0.87 | 0.89 | 0.83 | 0.78 | 0.76 | 0.66 | 0.85 | 0.72 | 0.92 | 0.71 | 0.62 | 0.72 | 0.73 |
| **A8Z** | 0.82 | 0.91 | 0.81 | 1 | 0.78 | 0.88 | 0.89 | 0.85 | 0.84 | 0.97 | 0.87 | 0.88 | 0.9 | 0.97 | 0.86 | 0.81 | 0.73 | 0.98 |
| **A9** | 0.76 | 0.82 | 0.8 | 0.76 | 1 | 0.75 | 0.8 | 0.78 | 0.78 | 0.78 | 0.69 | 0.77 | 0.74 | 0.67 | 0.78 | 0.62 | 0.76 | 0.69 |
| **AQ** | 0.88 | 0.87 | 0.9 | 0.87 | 0.77 | 1 | 0.94 | 0.88 | 0.86 | 0.97 | 0.87 | 0.97 | 0.84 | 1 | 0.81 | 0.81 | 0.75 | 0.94 |
| **AZ** | 0.89 | 0.88 | 0.89 | 0.88 | 0.8 | 0.94 | 1 | 0.91 | 0.9 | 0.97 | 0.8 | 0.97 | 0.82 | 0.99 | 0.82 | 0.75 | 0.84 | 0.93 |
| **A7** | 0.87 | 0.87 | 0.86 | 0.83 | 0.82 | 0.88 | 0.91 | 1 | 0.89 | 0.98 | 0.68 | 0.91 | 0.81 | 0.99 | 0.8 | 0.63 | 0.74 | 0.96 |
| **A6P** | 0.87 | 0.86 | 0.79 | 0.82 | 0.77 | 0.87 | 0.92 | 0.89 | 1 | 0.89 | 0.77 | 0.9 | 0.83 | 0.75 | 0.82 | 0.8 | 0.78 | 0.89 |
| **AJ3** | 0.99 | 0.97 | 0.76 | 0.97 | 0.75 | 0.97 | 0.98 | 0.98 | 0.89 | 1 | 0.72 | 0.88 | 0.91 | 0.76 | 0.82 | 0.77 | 0.81 | 0.89 |
| **AP7** | 0.71 | 0.85 | 0.63 | 0.88 | 0.65 | 0.86 | 0.86 | 0.74 | 0.77 | 0.71 | 1 | 0.75 | 0.81 | 0.54 | 0.8 | 0.83 | 0.75 | 0.8 |
| **AQV** | 0.9 | 0.89 | 0.86 | 0.86 | 0.75 | 0.96 | 0.96 | 0.91 | 0.88 | 0.88 | 0.77 | 1 | 0.85 | 0.77 | 0.8 | 0.76 | 0.82 | 0.85 |
| **ARF** | 0.82 | 0.9 | 0.72 | 0.91 | 0.74 | 0.86 | 0.85 | 0.83 | 0.85 | 0.9 | 0.8 | 0.84 | 1 | 0.74 | 0.82 | 0.81 | 0.81 | 0.92 |
| **AW6** | 0.96 | 0.94 | 0.92 | 0.94 | 0.65 | 1 | 0.99 | 0.98 | 0.72 | 0.72 | 0.5 | 0.73 | 0.7 | 1 | 0.69 | 0.51 | 0.49 | 0.7 |
| **AWF** | 0.8 | 0.88 | 0.7 | 0.86 | 0.76 | 0.82 | 0.82 | 0.81 | 0.82 | 0.82 | 0.81 | 0.8 | 0.83 | 0.73 | 1 | 0.81 | 0.77 | 0.86 |
| **AY9N** | 0.62 | 0.76 | 0.59 | 0.81 | 0.6 | 0.79 | 0.75 | 0.65 | 0.78 | 0.77 | 0.81 | 0.74 | 0.8 | 0.55 | 0.8 | 1 | 0.72 | 0.8 |
| **AY93** | 0.76 | 0.76 | 0.67 | 0.65 | 0.72 | 0.69 | 0.74 | 0.65 | 0.78 | 0.84 | 0.75 | 0.8 | 0.81 | 0.55 | 0.77 | 0.75 | 1 | 0.82 |
| **AYG** | 0.99 | 0.99 | 0.72 | 0.99 | 0.67 | 0.95 | 0.95 | 0.98 | 0.87 | 0.87 | 0.78 | 0.83 | 0.9 | 0.75 | 0.84 | 0.79 | 0.76 | 1 |

### 2.2.2 Function parameter optimisation and introduction of PM limit

Analysis of the results obtained from direct application of the original function showed that the conditions set out by it were

too strict to apply to the sensor data given the variations that can occur in AQS measurements, especially bearing in mind that the entire sensor network could be within the original single grid cell size analysed by Piersanti et al. (2015). Moreover, the high hourly PM$_{2.5}$ variation and very localised effects exhibited in a typical Irish winter PM$_{2.5}$ profile is not suited to the original function (Byrne et al., 2023). Thus, a PM mass concentration limit, $PM_{lim}$, was introduced to the function, with different relative concentration limits for the upper and lower PM values, $C_{lim, upper}$ and $C_{lim, lower}$, respectively. Treating larger

and smaller PM$_{2.5}$ values when assessing the similarity between two data series is useful in capturing the nuanced relationships and patterns in the data. It allows for the real-world significance of the data to be reflected, acknowledging the varying implications of PM$_{2.5}$ measurements based on the magnitude. Higher PM$_{2.5}$ values can indicate a pollution episode or specific local pollution sources, while lower values can represent background levels. Therefore, treating lower PM$_{2.5}$ values





with more leniency in the similarity assessment recognises that minor fluctuations in low hourly concentrations might not be

as concerning as similar deviations in higher concentrations and the health-related considerations associated with these high

concentrations.

Another potential advantage of the PM limit concerns the varying degrees of accuracy of the AQS measurements. Allowing the leeway introduced here in assessing the similarity of lesser measurement values considers potential measurement uncertainties with these devices. However, it is important to note that this approach is not accommodating sensor limitations

at the expense of accuracy but rather it is a strategy to ensure that the assessment remains faithful to the underlying air quality dynamics while accounting for the potential deficiencies in measurement equipment.

The differentiation between higher and lower PM values in the concentration similarity assessment is a strategic choice which acknowledges the complexity of $PM_{2.5}$ data, the varying significance of concentration levels, and the limitations of sensors. It allows for a more accurate representation of similarities while considering real world implications and

measurement uncertainties and minimises the potential biases that could arise from an indiscriminate approach, thus ensuring an impartial and unbiased evaluation.

When the function is applied to a pair of sensors, the resulting CSI can differ slightly depending on which sensor was classified as sensor A or sensor B in Equation (1) when computing the difference at each timestep. Due to the nature of the function, the denominator value of the relative difference calculation, the concentration of sensor A at a given timestep, is

what makes the difference. To counteract this and to avoid the possibility of large discrepancies between the CSI values for a sensor pair depending on which sensor is taken as A or B, the function was modified to have the geometric mean, or the square root of the product, of $C(A, t_i)$ and $C(B, t_i)$ used as the denominator. This ensured symmetry in the function so that the CSI values were identical regardless of which sensor was classified as A or B in a sensor pair.

Equation (2) shows the next form of the concentration similarity function (function notation has been modified to be more

suitable for this application).

$$CSI_{A,B} = \frac{\sum_{i=1}^{N_t} f}{N_t}, where\ f = \begin{cases} 1\ if\ \dfrac{|C(B,t_i) - C(A,\ t_i)|}{\sqrt{C(A,\ t_i) \times C(B,t_i)}}\ < 0.2\ and\ C(A,t_i)\ or\ C(B,t_i) > 15\ \mu g\ m^{-3} \\[2ex] 0\ if\ \dfrac{|C(B,t_i) - C(A,\ t_i)|}{\sqrt{C(A,\ t_i) \times C(B,t_i)}}\ > 0.2\ and\ C(A,t_i)\ or\ C(B,t_i) < 15\ \mu g\ m^{-3} \\[2ex] 1\ if\ \dfrac{|C(B,t_i) - C(A,\ t_i)|}{\sqrt{C(A,\ t_i) \times C(B,t_i)}}\ < 0.7\ and\ C(A,t_i)\ or\ C(B,t_i) > 15\ \mu g\ m^{-3} \\[2ex] 0\ if\ \dfrac{|C(B,t_i) - C(A,\ t_i)|}{\sqrt{C(A,\ t_i) \times C(B,t_i)}}\ > 0.7\ and\ C(A,t_i)\ or\ C(B,t_i) < 15\ \mu g\ m^{-3} \end{cases}$$

(2)

Where $C(A, t_i)$ and $C(B, t_i)$ are the $PM_{2.5}$ measurements from devices $A$ and $B$ at time, $t_i$.



### 2.2.3 Development and testing of the modified equation

The PM limit and associated concentration similarity limits introduced were chosen by iteratively testing the similarity function on the co-location data using different limits. Each co-located sensor pair was tested with different $PM_{lim}$ values (5, 10, 15, 20 µg m$^{-3}$) and with $C_{lim}$ values ranging from 0.1 to 2.0 in steps of 0.1 for both the upper and lower limits. This produced a $C_{lim}$ vs CSI comparison for each A-B pair for data above and below the corresponding $PM_{lim}$ value. It was clear that larger PM value comparisons (> 15 µg m$^{-3}$) tended to produce higher CSI values than lower PM values as expected. The

$C_{lim}$ value for each sensor comparison which gave a minimum CSI value of 0.95 was recorded with the overall mean of these $C_{lim}$ values above and below each $PM_{lim}$ value taken forward. The mean $C_{lim}$ pair values were then applied to the co-location measurements with the respective $PM_{lim}$ values to give final CSI values for each sensor pair, highlighting how the PM$_{2.5}$ concentration profile of each sensor compares to that of all the other sensors. The highest mean CSI value for all A-B pairs was found for $PM_{lim} = 15$ µg m$^{-3}$, $C_{lim, upper} = 0.2$, and $C_{lim, lower} = 0.7$. When applying these new limits, all sensor pairs gave

CSI > 0.85, with 99 % of pairs above 0.90 with an overall CSI mean of 0.98. These final limits enabled a good comparison for the hourly co-located AQS measurements (Table 2).

The CSI function was also applied to data obtained from the four co-located PurpleAir devices in order to make sure that the function was applicable across the two AQS types. The data was harmonised by following the same procedure as the Clarity data, through scaling each data from each sensor to the mean data series of all four sensors. Although this co-location period

was shorter than that of the Clarity dataset used for the function development, it still allowed for the CSI to be calculated from around 250 common data points per sensor pair. All device pairs reported a CSI close to 1.0, with a mean CSI of 0.99 (Table S3).





**Table 2: Concentration Similarity Indices for hourly averaged PM$_{2.5}$ measured by a range of co-located Clarity Node-S devices.** $PM_{lim}$ = 15 µg m$^{-3}$, $C_{lim, upper}$ = 0.2, $C_{lim, lower}$ = 0.7.

| | A3 | A4 | A8H | A8Z | A9 | AQ | AZ | A7 | A6P | AJ3 | AP7 | AQV | ARF | AW6 | AWF | AY9N | AY93 | AYG |
|---|---|---|---|---|---|---|---|---|---|---|---|---|---|---|---|---|---|---|
| **A3** | 1 | 0.97 | 0.99 | 0.96 | 0.92 | 0.99 | 1 | 0.99 | 0.99 | 1 | 0.96 | 1 | 0.97 | 1 | 0.96 | 0.94 | 0.99 | 1 |
| **A4** | | 1 | 0.97 | 0.99 | 0.95 | 0.98 | 0.99 | 0.99 | 0.99 | 1 | 0.99 | 0.99 | 0.99 | 1 | 0.98 | 0.97 | 0.97 | 1 |
| **A8H** | | | 1 | 0.96 | 0.92 | 0.99 | 1 | 0.99 | 0.99 | 0.97 | 0.94 | 0.99 | 0.96 | 0.98 | 0.96 | 0.94 | 0.97 | 0.98 |
| **A8Z** | | | | 1 | 0.94 | 0.97 | 0.99 | 0.98 | 0.98 | 1 | 0.98 | 0.98 | 1 | 1 | 0.98 | 0.96 | 0.97 | 1 |
| **A9** | | | | | 1 | 0.92 | 0.92 | 0.94 | 0.95 | 0.97 | 0.96 | 0.95 | 0.97 | 0.92 | 0.97 | 0.95 | 0.96 | 0.96 |
| **AQ** | | | | | | 1 | 1 | 1 | 0.99 | 1 | 0.97 | 1 | 0.97 | 1 | 0.96 | 0.97 | 1 | 1 |
| **AZ** | | | | | | | 1 | 0.99 | 0.99 | 1 | 0.97 | 1 | 0.98 | 1 | 0.96 | 0.95 | 1 | 1 |
| **A7** | | | | | | | | 1 | 0.99 | 1 | 0.95 | 0.99 | 0.98 | 1 | 0.96 | 0.94 | 0.97 | 1 |
| **A6P** | | | | | | | | | 1 | 0.99 | 0.97 | 0.99 | 0.99 | 0.96 | 0.97 | 0.97 | 0.99 | 1 |
| **AJ3** | | | | | | | | | | 1 | 0.98 | 0.98 | 0.99 | 0.95 | 0.99 | 0.98 | 1 | 0.99 |
| **AP7** | | | | | | | | | | | 1 | 0.96 | 0.99 | 0.85 | 0.99 | 0.99 | 0.97 | 0.98 |
| **AQV** | | | | | | | | | | | | 1 | 0.98 | 0.98 | 0.97 | 0.96 | 0.99 | 0.99 |
| **ARF** | | | | | | | | | | | | | 1 | 0.96 | 0.98 | 0.99 | 0.99 | 1 |
| **AW6** | | | | | | | | | | | | | | 1 | 0.92 | 0.87 | 0.9 | 0.97 |
| **AWF** | | | | | | | | | | | | | | | 1 | 0.98 | 0.97 | 0.98 |
| **AY9N** | | | | | | | | | | | | | | | | 1 | 0.97 | 0.99 |
| **AY93** | | | | | | | | | | | | | | | | | 1 | 0.99 |
| **AYG** | | | | | | | | | | | | | | | | | | 1 |

The function described in Eq. 2 was further tested by comparing one sensor, A6P, to numerous sets of synthetic data created from that sensor's measurements to assess the impact of a range of scenarios. Comparing the A6P data to itself establishes a baseline for the comparison where the CSI is 1 and any subsequent adjustments to the data to create the synthetic data can be

explored, resulting in a new CSI. The first scenario investigated changes in CSI when outliers are present in the data. To explore this, the A6P data was changed so a certain number of data points could be considered outliers (n = 1, 10, 500, 1000). To classify a data point as an outlier, the selected data point was increased by 100 µg m$^{-3}$ in order to ensure discrepancy between it and the original value. The function was then tested in a scenario where the mean remained constant

but the variance of the data was increased, and it was also tested in a scenario where the entire data was merely offset by 5, 10, 15, and 20 µg m$^{-3}$. The effects of these tests on the CSI results for A6P are shown in Table 3. It is clear that in the case of data with higher variability but the same overall mean, the CSI is impacted, because even when the variance is increased by just a factor of 1.5 the CSI is significantly reduced, indicating that such a dataset is dissimilar to the original. Offsetting the





data by different degrees also shows a major effect which means that such a dataset is deemed dissimilar by the method.
However, the method is quite robust with respect to outliers. As the method yields a time-averaged result, low numbers of outliers do not hugely affect the index for a given sensor pair. So, two datasets that are generally similar, but where one experiences some outliers, will be deemed similar by the method. The development and analysis of the similarity index function in this way provided a basis for what to consider when applying the function to the field data.

**Table 3: Influence of data outliers and other factors on CSI determined in test scenarios.**

| Number of outliers | CSI | Standard Deviation factor increase ($\mu$g m$^{-3}$) | CSI | PM$_{2.5}$ positive offset ($\mu$g m$^{-3}$) | CSI | PM$_{2.5}$ negative offset ($\mu$g m$^{-3}$) | CSI |
|---|---|---|---|---|---|---|---|
| 0 | 1 | 0 | 1 | 0 | 1 | 0 | 1 |
| 1 | 1 | 1.5 | 0.52 | 5 | 0.54 | 5 | 0.34 |
| 10 | 1 | 2 | 0.29 | 10 | 0.05 | 10 | 0.02 |
| 500 | 0.89 | 4 | 0.10 | 15 | 0.01 | 15 | 0.01 |
| 1000 | 0.77 | | | 20 | 0.01 | 20 | 0.004 |

## 260 2.3 Application to sensor networks and analysis of spatial trends

The CSI methodology developed above was subsequently applied to the Dungarvan and Cork City sensor networks to evaluate the similarity and spatial variations in PM$_{2.5}$. A systematic pairwise comparison approach was employed, wherein each sensor was individually compared to every other sensor within the network.

Hierarchical clustering and fuzzy $c$-means (FCM) clustering were both performed on the CSI results, to identify groupings
based on each sensors' relationship to other sensors in the network which can then be reflected spatially. Cluster analysis is a valuable unsupervised analysis technique used to identify natural groupings in a dataset by classifying the data into distinct groups, or clusters, without needing pre classified or labelled data to train the algorithm. It systematically works to separate the data by minimising within group variation and maximizing between group variation. Cluster analysis is often used in air quality analysis, including describing pollution diurnal variation, identifying distinct diurnal patterns, pollution source
identification, and identifying spatial patterns in particle compositions (Austin et al., 2012; Flemming et al., 2005; Austin et al., 2013).

Hierarchical clustering does not separate the data into a defined number of clusters in a single step, but rather consists of a series of separations which typically goes from a single cluster containing all of the data, to $n$ clusters each containing an individual sample (when the data is an $n \times m$ matrix for $n$ samples and $m$ data points in each sample) (Everitt et al., 2011).
The procedure typically includes a dendrogram showing the tree-like structure of the nested clusters. This type of clustering gives an advantage over partition-based algorithms, whereby the user is not required to specify the number of clusters.





Fuzzy c-means clustering is an example of a partitional clustering technique, where the number of clusters must be predefined. However, another distinctive feature separating it from hierarchical clustering is that it is a soft clustering method. In hard (i.e. non-fuzzy) clustering, each point belongs exclusively to a single cluster, whereas in soft clustering, the output is a membership score or probability likelihood of a data point belonging to each of the pre-defined clusters (Gentle et al., 1991). The assignment of a member to a group is a distribution over all available clusters. The partition that gives the closest hard clustering to the fuzzy output can be obtained by assigning each object to the cluster in which it has the largest membership score. However, the information achieved with soft clustering can be particularly useful when dealing with datasets exhibiting overlapping patterns or uncertainties in classification as opposed to directly partitioning into hard clusters (Gentle et al., 1991).

With both clustering techniques, the quality of cluster assignments can be assessed with various evaluation metrics to choose the optimal number of clusters. As the "true" cluster classifications are not known here, validation must be performed using the clustering algorithm itself. To assess the quality of the hierarchical clustering assignments, the Silhouette metric was used along with the Calinski-Harabasz index to assess the FCM assignments (Caliñski and Harabasz, 1974; Rousseeuw, 1987). The Silhouette score, ranging from -1 to +1, can be calculated for each member of a cluster and then the mean Silhouette score from all members indicates an overall assignment quality for members of that cluster, with a high score closer to 1 indicating higher quality clusters, and a low or negative score indicating poorer cluster assignments. The Calinski-Harabasz index also quantifies the quality of cluster assignments with higher scores indicating better quality. The metrics were used to test for the optimal number of clusters for each algorithm.

Both clustering approaches were selected to provide further understanding of the inherent spatial structures concealed within the CSI results. Hierarchical clustering offers the hierarchical representation of clusters, aiding in the identification of nested relationships, while FCM allows for a more flexible approach to the cluster assignments when the number of clusters is not known a priori.

## 3. Results and Discussion

### 3.1 Dungarvan PM$_{2.5}$ sensor network

Analysis of the harmonised data obtained from the sensors in the Dungarvan PM$_{2.5}$ network was conducted to determine CSI values and assess the spatial variation of air pollution across the town. Although the PM$_{2.5}$ concentrations are not as accurate as those collected by reference instrumentation, any relative differences between the sensors and individual sensor data trends can be regarded as genuine due to the low inter-sensor variation observed after data harmonisation procedures.

The temporal and spatial trends of PM$_{2.5}$ across the Dungarvan sensor network are reflected in the average diurnal plots obtained for each sensor, Fig. 1. These diurnal profiles all show large evening peaks in PM$_{2.5}$, which are typical for towns and cities in Ireland affected by residential solid fuel burning during winter evenings (Healy et al., 2010; Dall'Osto et al., 2014; Wenger et al., 2020). However, there are clear disparities in some of the average evening peak values between the





sensors. One group of sensors has maximum values above 35 µg m$^{-3}$ (A3, A4, A8H, A9, AQ, A7, AW6, AQV), while the

sensors with maxima below 35 µg m$^{-3}$ can be further divided into three smaller groups. Sensors labelled AJ3, AWF, and AZ all have a maximum PM$_{2.5}$ concentration around 30 µg m$^{-3}$; sensors AY9N, AY93, ARF, A8Z, AYG, and A6P all have maxima in the 20-26 µg m$^{-3}$ range, while AP7 has a significantly lower evening peak than all other devices.

Most sensors exhibited the diurnal maximum around the same time of day, between 18:00 and 20:00, however AP7 and ARF, showed a slightly delayed peak from 20:00 to 22:00. AP7 had the lowest peak concentration and did not exhibit the

sharp rise and subsequent decrease associated with evening solid fuel burning that the other sensors showed. AP7 was located on the south-western edge of the town and since the predominant wind direction is south westerly, did not experience as much local pollution as the other locations in the network.

Summary statistics obtained for the 18 sensors in the Dungarvan network are listed in Table 4. Unsurprisingly, most of the devices with diurnal maxima > 35 µg m$^{-3}$ have the highest mean, median, and maximum values. Out of this subset of

devices, AQV has the lowest overall mean (15 µg m$^{-3}$), but still has a relatively high standard deviation (22 µg m$^{-3}$), indicating the PM$_{2.5}$ values tend to vary widely but are lower on average. This could be indicative of fluctuating particle concentrations, consistent with intermittent pollution sources such as residential solid fuel burning.

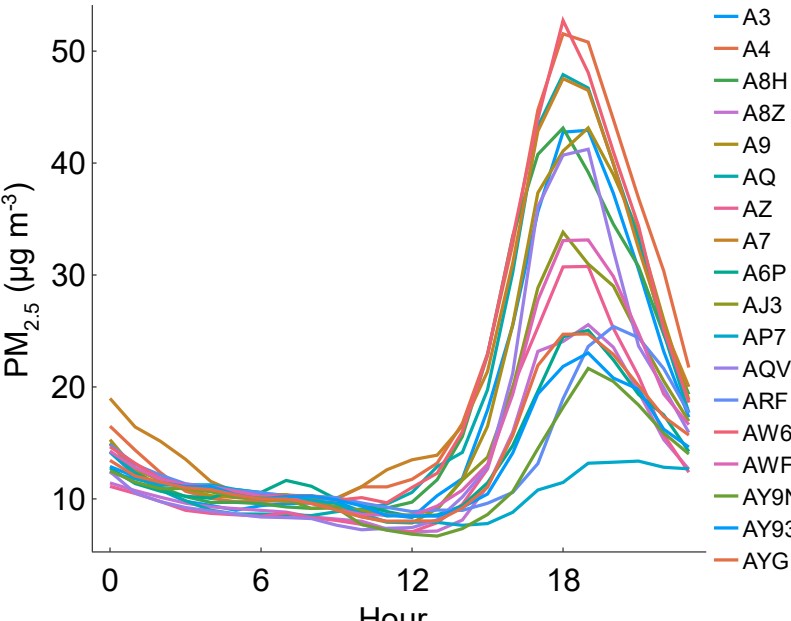

**Figure 1: Diurnal profiles for hourly averaged measurements of PM$_{2.5}$ in the Dungarvan sensor network (September 2022 to May**
**2023).**



**Table 4: Summary statistics of hourly average PM$_{2.5}$ concentrations obtained for all sensors in the Dungarvan sensor network (September 2022 to May 2023).**

| ID | Mean | Median | Standard Deviation | Maximum hourly value | Maximum diurnal value | Hour of maximum diurnal value |
|---|---|---|---|---|---|---|
| | µg m$^{-3}$ | µg m$^{-3}$ | µg m$^{-3}$ | µg m$^{-3}$ | µg m$^{-3}$ | |
| AP7 | 11 | 7 | 12 | 153 | 12 | 21 |
| AY9N | 12 | 7 | 14 | 136 | 23 | 19 |
| A8Z | 13 | 7 | 16 | 274 | 25 | 19 |
| A6P | 13 | 8 | 18 | 311 | 26 | 19 |
| AY93 | 13 | 8 | 15 | 176 | 22 | 19 |
| AZ | 14 | 7 | 18 | 286 | 30 | 19 |
| ARF | 14 | 8 | 18 | 243 | 26 | 20 |
| AYG | 14 | 8 | 16 | 259 | 24 | 19 |
| AQV | 15 | 8 | 22 | 281 | 44 | 19 |
| AJ3 | 16 | 9 | 19 | 270 | 33 | 18 |
| AWF | 16 | 9 | 17 | 189 | 31 | 19 |
| A3 | 18 | 9 | 27 | 412 | 45 | 19 |
| A9 | 18 | 9 | 27 | 409 | 45 | 19 |
| A8H | 19 | 9 | 28 | 482 | 40 | 18 |
| AQ | 19 | 10 | 28 | 480 | 48 | 18 |
| A4 | 21 | 12 | 27 | 370 | 52 | 18 |
| A7 | 21 | 12 | 27 | 361 | 45 | 18 |
| AW6 | 21 | 11 | 26 | 319 | 51 | 18 |

### 3.1.1 Concentration Similarity Index

The matrix of CSI values obtained for the Dungarvan sensor network is shown in Table 5. The results can be analysed in a number of ways. Firstly, the indices for one sensor can be used to assess how similar or dissimilar the measurements are to all other sensors in the network, thus providing information on the spatial representativeness of that particular location. Secondly, the indices of all sensors can be looked at together to elucidate any potential relationship between sensor measurement locations.

The minimum CSI value (0.85) determined during the co-location deployment can act as the lower limit for when two sensor locations can be considered very similar. The reported CSI values for Dungarvan sensors ranged from 0.48 (ARF vs A7) to 0.79 (AYG vs AWF) with a mean of 0.61, indicating a significant difference in air quality representation between locations





across the town. The device with the lowest mean of its CSI values with respect to the other locations was A4 (0.55), and although device ARF was only slightly above this (0.57), it reported a larger range of CSI values, including the lowest of the
entire dataset. AJ3, AQV, and AYG all shared the highest mean CSI values (0.66).

To further investigate the effect of solid fuel burning on local air quality, the CSI function was applied to data from two isolated months – January and May 2023. The purpose of this assessment was to evaluate the extent to which residential solid fuel burning dictates the CSI between two sensors, given that one month (January) will have higher $PM_{2.5}$ levels with measurements heavily influenced by solid fuel burning, and the other will not (May). For both months, all sensors had data
capture above 65 % and the mean capture was 94 % for January and 92 % for May. The January mean CSI from all comparisons was 0.51, and the May mean CSI was 0.84 (Table S4, Table S5). The large discrepancy between the mean CSI for January and May is most likely due to the higher variation typically seen in wintertime $PM_{2.5}$ due to residential solid fuel burning. This highlights the importance of seasonality when assessing the spatial representativeness of monitoring network locations.






**Table 5: Concentration Similarity Indices for the hourly averaged PM$_{2.5}$ concentrations measured by Clarity Node-S devices in the Dungarvan sensor network.**

| | A3 | A4 | A8H | A8Z | A9 | AQ | AZ | A7 | A6P | AJ3 | AP7 | AQV | ARF | AW6 | AWF | AY9N | AY93 | AYG |
|---|---|---|---|---|---|---|---|---|---|---|---|---|---|---|---|---|---|---|
| A3 | 1 | 0.56 | 0.64 | 0.56 | 0.58 | 0.67 | 0.58 | 0.62 | 0.64 | 0.59 | 0.53 | 0.61 | 0.5 | 0.63 | 0.55 | 0.59 | 0.6 | 0.57 |
| A4 | | 1 | 0.53 | 0.52 | 0.55 | 0.53 | 0.56 | 0.58 | 0.53 | 0.57 | 0.55 | 0.58 | 0.49 | 0.54 | 0.55 | 0.56 | 0.55 | 0.55 |
| A8H | | | 1 | 0.58 | 0.6 | 0.61 | 0.59 | 0.57 | 0.61 | 0.61 | 0.57 | 0.64 | 0.54 | 0.61 | 0.56 | 0.61 | 0.62 | 0.61 |
| A8Z | | | | 1 | 0.62 | 0.55 | 0.69 | 0.54 | 0.62 | 0.65 | 0.71 | 0.66 | 0.6 | 0.56 | 0.62 | 0.66 | 0.62 | 0.67 |
| A9 | | | | | 1 | 0.53 | 0.6 | 0.57 | 0.58 | 0.63 | 0.6 | 0.63 | 0.52 | 0.58 | 0.6 | 0.61 | 0.58 | 0.6 |
| AQ | | | | | | 1 | 0.58 | 0.57 | 0.6 | 0.59 | 0.5 | 0.62 | 0.49 | 0.63 | 0.53 | 0.56 | 0.58 | 0.56 |
| AZ | | | | | | | 1 | 0.58 | 0.63 | 0.68 | 0.68 | 0.71 | 0.6 | 0.62 | 0.66 | 0.62 | 0.61 | 0.74 |
| A7 | | | | | | | | 1 | 0.57 | 0.58 | 0.53 | 0.6 | 0.48 | 0.62 | 0.55 | 0.56 | 0.57 | 0.55 |
| A6P | | | | | | | | | 1 | 0.72 | 0.62 | 0.66 | 0.58 | 0.63 | 0.64 | 0.65 | 0.68 | 0.71 |
| AJ3 | | | | | | | | | | 1 | 0.64 | 0.76 | 0.6 | 0.64 | 0.76 | 0.67 | 0.7 | 0.78 |
| AP7 | | | | | | | | | | | 1 | 0.64 | 0.67 | 0.52 | 0.64 | 0.65 | 0.63 | 0.69 |
| AQV | | | | | | | | | | | | 1 | 0.59 | 0.65 | 0.72 | 0.65 | 0.68 | 0.77 |
| ARF | | | | | | | | | | | | | 1 | 0.56 | 0.59 | 0.62 | 0.61 | 0.63 |
| AW6 | | | | | | | | | | | | | | 1 | 0.61 | 0.6 | 0.62 | 0.62 |
| AWF | | | | | | | | | | | | | | | 1 | 0.66 | 0.68 | 0.79 |
| AY9N | | | | | | | | | | | | | | | | 1 | 0.59 | 0.67 |
| AY93 | | | | | | | | | | | | | | | | | 1 | 0.72 |
| AYG | | | | | | | | | | | | | | | | | | 1 |

### 3.1.2 Clustering

Clustering techniques were employed on the CSI matrix to uncover any inherent spatial relationships between different
locations in the network. Hierarchical clustering produced a dendrogram showing the hierarchical relationship between the
sensor locations and was used to identify clusters (Figure 2). The highest mean Silhouette score was found with 2 clusters





(Figure S4). However, it was not a high Silhouette score (0.19), indicating that the quality of the cluster assignments was low. The highest Calinski-Harabasz index corresponded to the assignment of members to 2 clusters when applying the FCM clustering (Figure S5).

From both the dendrogram (Figure 2) and the FCM membership weights (Figure 3), it is clear that devices A4 through to AQ are grouped together in one cluster (Cluster 1), and devices AQV to AP7 are grouped in another cluster (Cluster 2). This split is very similar to the easily visualised groupings shown in the diurnal profile maxima (Figure 1), with the only difference being device AQV. The devices in Cluster 1 are also those with the highest mean $PM_{2.5}$ for the measurement period. The mean CSI for each sensor mostly corresponds to the cluster assignments, with Cluster 1 devices having a mean CSI equal to

or below 0.6, and all devices in Cluster 2 have a mean CSI above 0.6, except for device ARF. Interestingly, this grouping also appears to have spatial importance too, as shown in Fig. 4. Cluster 2 devices are mainly located around the edge of the town and generally experience cleaner air, while Cluster 1 devices are located in central and residential areas, which are more polluted during winter months.

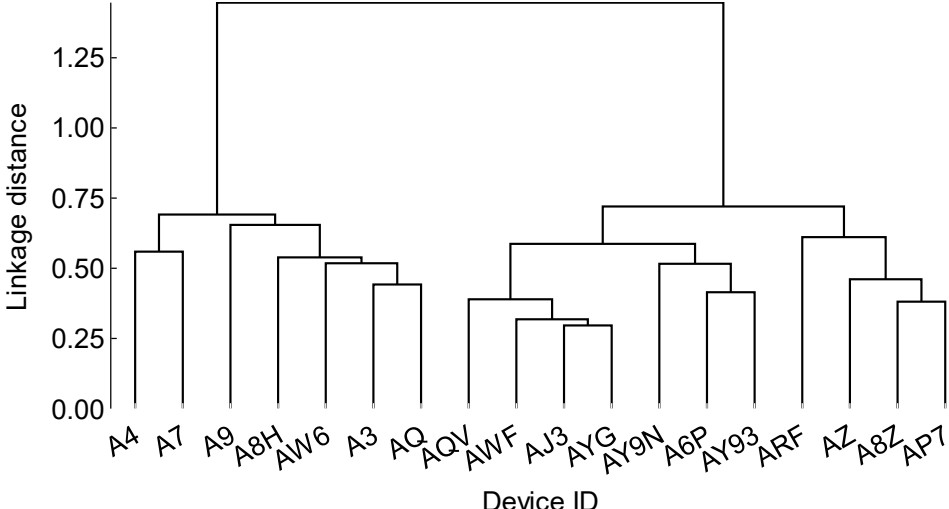

**Figure 2: Dendrogram output from hierarchical clustering of the CSI data from the Dungarvan sensor network.**





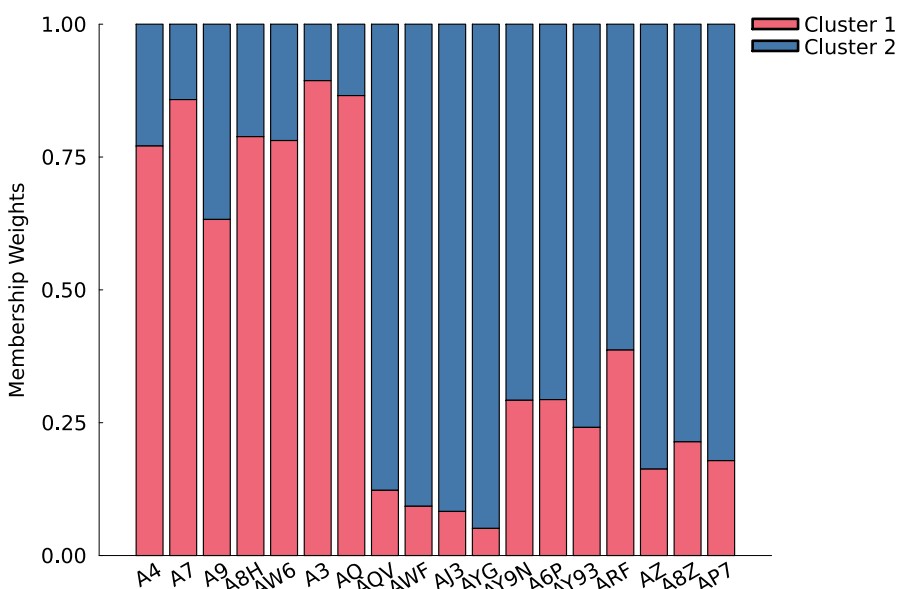

**Figure 3: Membership weights from FCM clustering of the CSI data from the Dungarvan sensor network.**

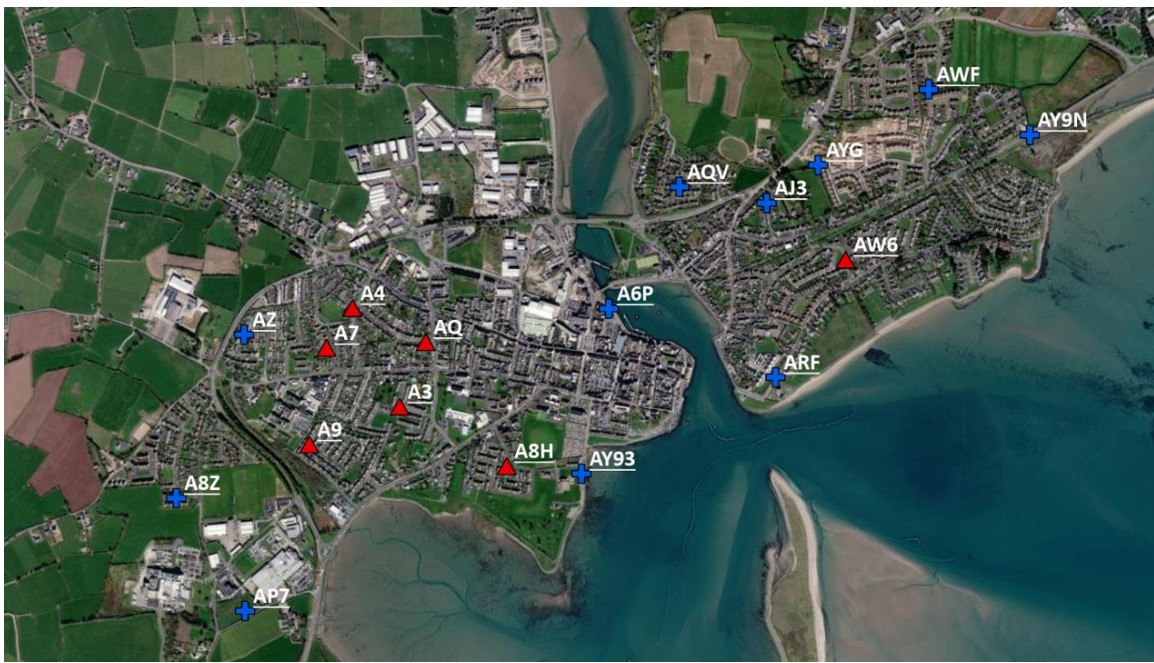

**Figure 4: Dungarvan AQS locations with two cluster groups indicated. Cluster 1 devices (red triangle markers) are mainly located in central and residential areas, while cluster 2 devices (blue cross markers) are mainly located on the edge of the town. (Map obtained from Esri, DigitalGlobe, GeoEye, i-cubed, USDA FSA, USGS, AEX, Getmapping, Aerogrid, IGN, IGP, swisstopo, and the GIS User Community)**





### 3.2 Cork City PM$_{2.5}$ sensor network

The same approach was used to analyse the data collected by the Cork City AQS network. In this case, the corrected measurements are indicative of the actual PM$_{2.5}$ experienced in each location. The diurnal plots for each sensor in the Cork City network are similar to those observed in Dungarvan, with a sizeable evening peak in PM$_{2.5}$ concentrations (19:00-21:00) due to emissions from residential solid fuel burning. Again, there is considerable variation in the peak concentration of PM$_{2.5}$ (Figure 5). Device MTU showed the lowest diurnal average maximum of 9 µg m$^{-3}$. This device is located on the western side

of the city and has few upwind pollution sources contributing to air pollution at the location as the prevailing wind direction is from the South-West. Devices CCC12 and CCC9 both showed the highest diurnal average maximum, 17 µg m$^{-3}$. CCC12 is located northeast of the city, and so likely experiences urban PM$_{2.5}$ sources up-wind from it or has strong localised sources. Similarly, CCC9 is located to the east of the city, in a residential area. Table 6 contains summary statistics for each of the sensors in the Cork City network. Some devices had very high PM$_{2.5}$ maxima, e.g. 201 µg m$^{-3}$ for CCC11, which were

more than double the maxima of other devices, e.g., CCC8 which had the lowest overall maximum of 47 µg m$^{-3}$. Device MTU had the lowest diurnal maximum value, indicating that this location is the least affected by local emissions from solid fuel burning. However, it measured a significant overall PM$_{2.5}$ maximum of 99 µg m$^{-3}$ and significant spikes in pollution were occasionally observed, likely due to meteorological conditions or specific localised effects. When looking at all of the parameters listed in Table 6, CCC11 stands out. This sensor has the highest maximum hourly average PM$_{2.5}$ concentration in

the network, but the standard deviation (8 µg m$^{-3}$) is in the middle of the range, indicating that the location had relatively stable PM$_{2.5}$ levels throughout the measurement period with less variation than other devices but was still susceptible to occasional spikes in PM$_{2.5}$.

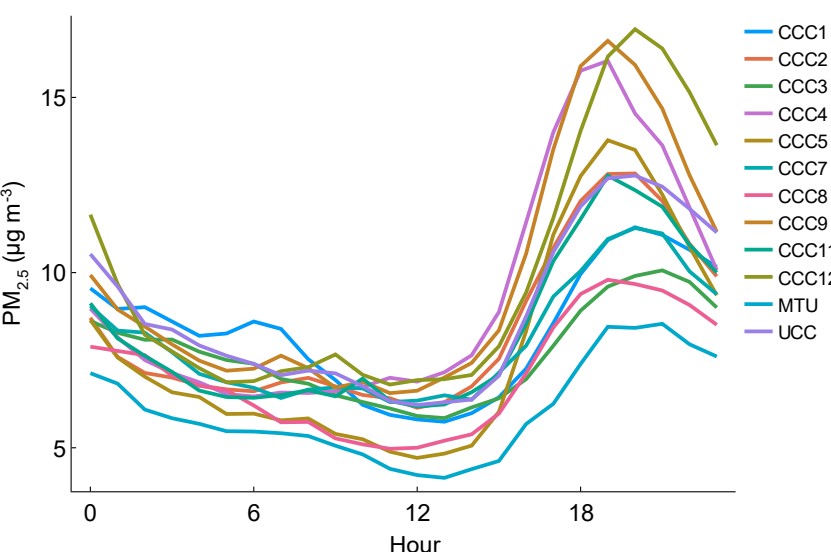

**Figure 5: Diurnal PM$_{2.5}$ profiles for all AQS in the Cork City network (January to May and September to December 2021).**





**Table 6: Summary statistics of hourly averaged PM$_{2.5}$ obtained for all sensors in the Cork City network (January to May and September to December 2021).**

| Device Label | Mean | Median | Standard Deviation | Maximum | Maximum diurnal value | Hour of maximum diurnal value |
|---|---|---|---|---|---|---|
| | µg m$^{-3}$ | µg m$^{-3}$ | µg m$^{-3}$ | µg m$^{-3}$ | µg m$^{-3}$ | |
| MTU | 6 | 4 | 6 | 99 | 9 | 21 |
| CCC8 | 7 | 5 | 6 | 47 | 10 | 19 |
| CCC3 | 8 | 5 | 7 | 61 | 10 | 21 |
| CCC5 | 8 | 5 | 10 | 181 | 14 | 19 |
| CCC | 8 | 6 | 7 | 71 | 11 | 20 |
| CCC11 | 8 | 6 | 8 | 201 | 13 | 19 |
| CCC1 | 8 | 6 | 8 | 92 | 11 | 20 |
| CCC2 | 8 | 6 | 8 | 122 | 13 | 20 |
| UCC | 9 | 6 | 8 | 108 | 13 | 20 |
| CCC4 | 9 | 7 | 8 | 97 | 16 | 19 |
| CCC9 | 10 | 7 | 10 | 158 | 17 | 19 |
| CCC12 | 10 | 7 | 10 | 117 | 17 | 20 |

### 3.2.1 Concentration Similarity Index

The matrix of CSI values obtained for the Cork City sensor network is shown in Table 7. The values range from 0.52 (CCC12 vs MTU and CCC9 vs MTU) to 0.85 (CCC2 vs CCC11) with a mean of 0.71. The high maximum CSI indicates a

high degree of similarity between those locations in the network, and overall, the Cork City locations show a higher degree of similarity compared to those in Dungarvan.

The isolated CSI results for the months of January and May 2021 were also assessed for Cork City. The average data coverage during both periods was 92 %. The mean CSI value in January (0.55) was considerably lower than that observed in May (0.82), Table S6, Table S7. This result is similar to that found for the Dungarvan network, again indicating that the

large difference in mean scores between the two months can be attributed to higher wintertime PM$_{2.5}$ variation by residential solid fuel burning.





**Table 7: Concentration Similarity Indices for the hourly averaged PM$_{2.5}$ concentrations measured by PurpleAir devices in the Cork City AQS network.**

|        | CCC1 | CCC2 | CCC3 | CCC4 | CCC5 | CCC7 | CCC8 | CCC9 | CCC11 | CCC12 | MTU  | UCC  |
|--------|------|------|------|------|------|------|------|------|-------|-------|------|------|
| CCC1   | 1    | 0.73 | 0.76 | 0.68 | 0.65 | 0.71 | 0.66 | 0.64 | 0.76  | 0.67  | 0.66 | 0.76 |
| CCC2   |      | 1    | 0.79 | 0.82 | 0.65 | 0.77 | 0.68 | 0.73 | 0.85  | 0.78  | 0.61 | 0.79 |
| CCC3   |      |      | 1    | 0.73 | 0.76 | 0.8  | 0.8  | 0.65 | 0.82  | 0.7   | 0.76 | 0.8  |
| CCC4   |      |      |      | 1    | 0.63 | 0.73 | 0.64 | 0.76 | 0.82  | 0.78  | 0.56 | 0.77 |
| CCC5   |      |      |      |      | 1    | 0.65 | 0.74 | 0.7  | 0.66  | 0.6   | 0.69 | 0.71 |
| CCC7   |      |      |      |      |      | 1    | 0.68 | 0.65 | 0.78  | 0.7   | 0.66 | 0.73 |
| CCC8   |      |      |      |      |      |      | 1    | 0.7  | 0.67  | 0.6   | 0.67 | 0.74 |
| CCC9   |      |      |      |      |      |      |      | 1    | 0.72  | 0.74  | 0.52 | 0.8  |
| CCC11  |      |      |      |      |      |      |      |      | 1     | 0.79  | 0.61 | 0.84 |
| CCC12  |      |      |      |      |      |      |      |      |       | 1     | 0.52 | 0.77 |
| MTU    |      |      |      |      |      |      |      |      |       |       | 1    | 0.62 |
| UCC    |      |      |      |      |      |      |      |      |       |       |      | 1    |

### 3.2.2 Clustering

The two clustering algorithms were applied to investigate the CSI results of the Cork City network. The Silhouette Scores for each number of assigned clusters (2 to 5) were low, with 2 clusters showing the highest mean score (Figure S6). Similarly, with the FCM analysis, 2 clusters showed the highest score with the Calinski-Harabasz indices (Figure S7).

The dendrogram produced from the hierarchical clustering and the membership weights for 2 clusters from FCM clustering are shown in Fig. 6 and Fig. 7, respectively. It is clear that devices MTU, CCC5, and CCC8 are all grouped together in one branch, Cluster 2, with the remainder of the devices in the other branch. The one assignment difference between the two clustering methods is CCC3, which has a higher membership weight towards Cluster 2 with the FCM method but does not branch with that cluster in the dendrogram. However, its membership weight is close to 0.5. CCC1 also shows a membership weight close to 0.5, however it is showing a higher weight towards Cluster 1, as per the hierarchical clustering results. Devices in Cluster 2, except for CCC3, all have the lowest mean CSI value.

Similar to the Dungarvan results, there appears to be a spatial component to the cluster groupings, with devices in Cluster 2 being mainly on the western side of the city, Fig. 8. Interestingly, device CCC7, located in a commuter town on the western side of the city boundary, is grouped in Cluster 1, along with devices mainly in urban residential type sites, instead of being grouped with other devices on the western edge of the city. This indicates it has a more comparable CSI profile to the urban residential sites than the locations closer to it, further emphasising the importance of location type over physical proximity.





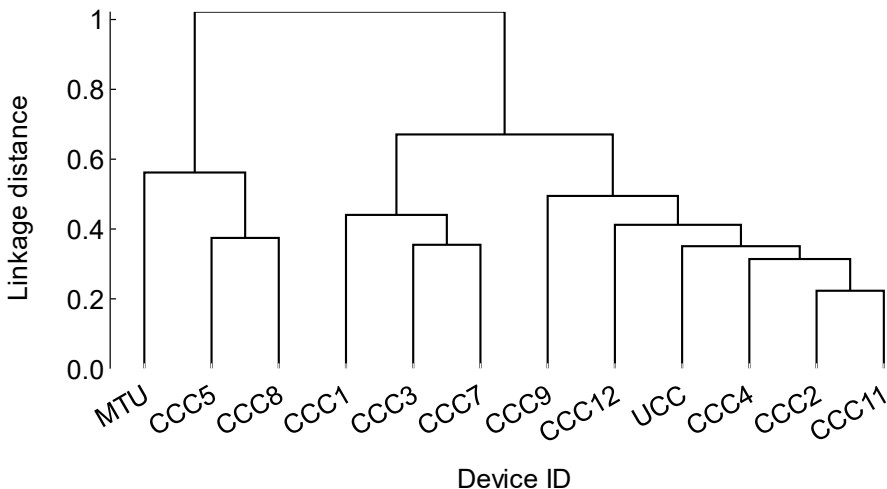

**Figure 6: Dendrogram output from hierarchical clustering of the CSI data from the Cork sensor network.**

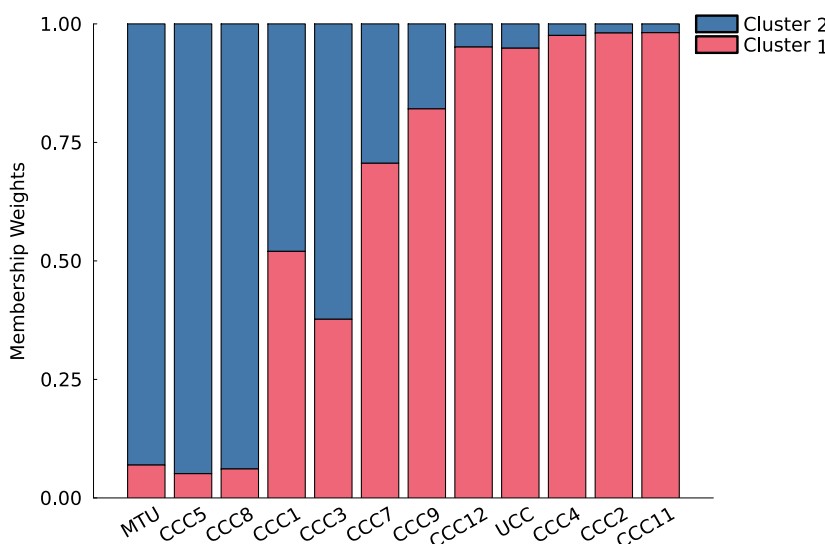

**Figure 7: Membership weights from FCM clustering of the CSI data from the Cork sensor network.**





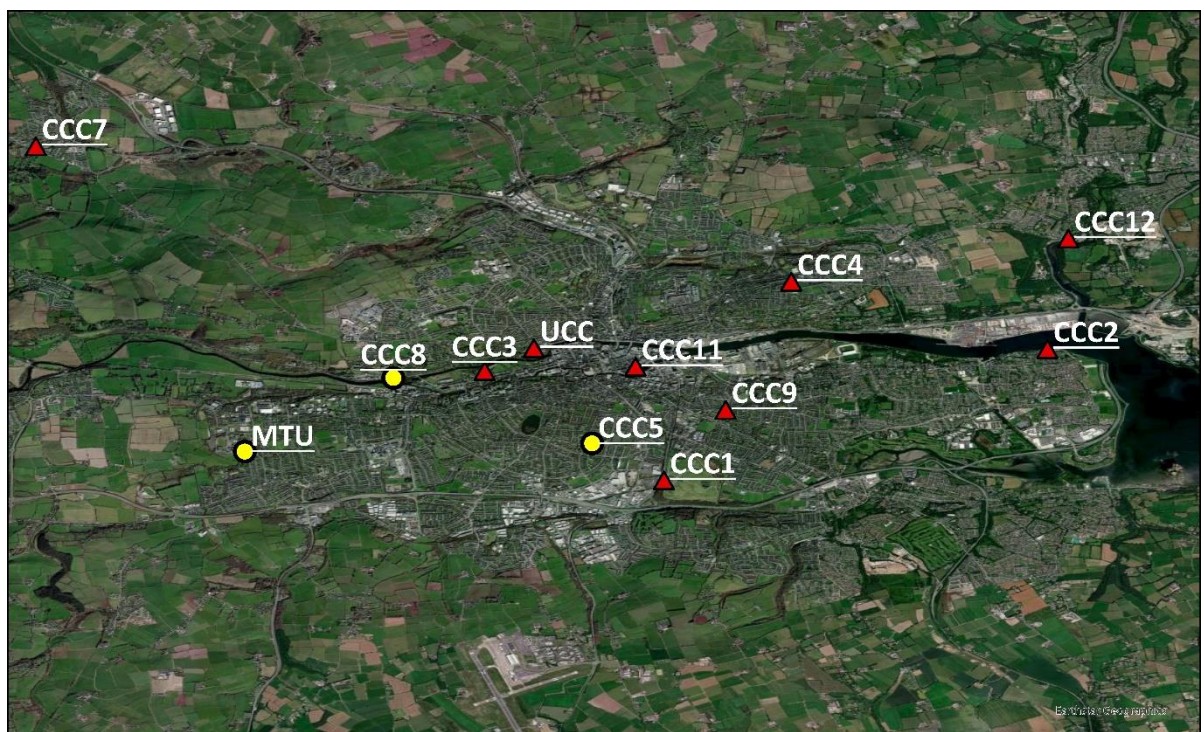

**Figure 8: Cock City AQS locations with 2 cluster groups indicated. Cluster 1 devices (red triangle markers) are located in the city centre and east/northeast, while Cluster 2 devices (yellow circle markers) are mainly located on the western side of the city. (Map obtained from Esri, DigitalGlobe, GeoEye, i-cubed, USDA FSA, USGS, AEX, Getmapping, Aerogrid, IGN, IGP, swisstopo, and the GIS User Community)**

### 3.3 Application of the CSI to assess representativeness of air quality monitoring locations

One key benefit of the CSI metric for AQS networks is that one sensor can be singled out and its overall degree of similarity to measurements from other locations can be determined. This analysis can be used to assess the spatial representativeness of a given location in the AQS network by quantitively exploring how similar its PM$_{2.5}$ profile is to other locations. If a network sensor is co-located with a reference instrument, then the CSI values for that sensor can be used to provide a measure of the representativeness of the designated monitoring location and how well it informs the assessment of population exposure to air pollution.

In Dungarvan, the device A6P was co-located with a PM$_{2.5}$ instrument (Osiris, Turnkey) deployed as part of the national air quality monitoring network. The instrument is not a reference instrument but is certified to provide indicative measurements of PM$_{2.5}$ (National Ambient Air Quality Monitoring Network , 2023; Osiris, 2024). The similarity indices for A6P are included in Table 5 and represented spatially in Fig. 9. All CSI values are below the minimum threshold of 0.85 for two Clarity S-node devices in the Dungarvan network to be considered very similar. The most similar devices are found to the north and south of this location, AQV and AY93. Interestingly, the similarity of PM profiles does not decrease with increasing distance from A6P. Devices on the furthest western (AZ, A8Z, AP7) and eastern (AWF, AY9N) edges of the





town are within 0.6 to 0.7, yet devices A4, A7, AQ, and A9 are all below 0.6 despite being physically closer to A6P. This

suggests that the location type is more important when it comes to assessing the similarity of locations within Irish towns as

opposed to physical proximity, as A4, A7, AQ, and A9 are all fully surrounded by residential areas, whereas the other

mentioned devices are in more open areas.

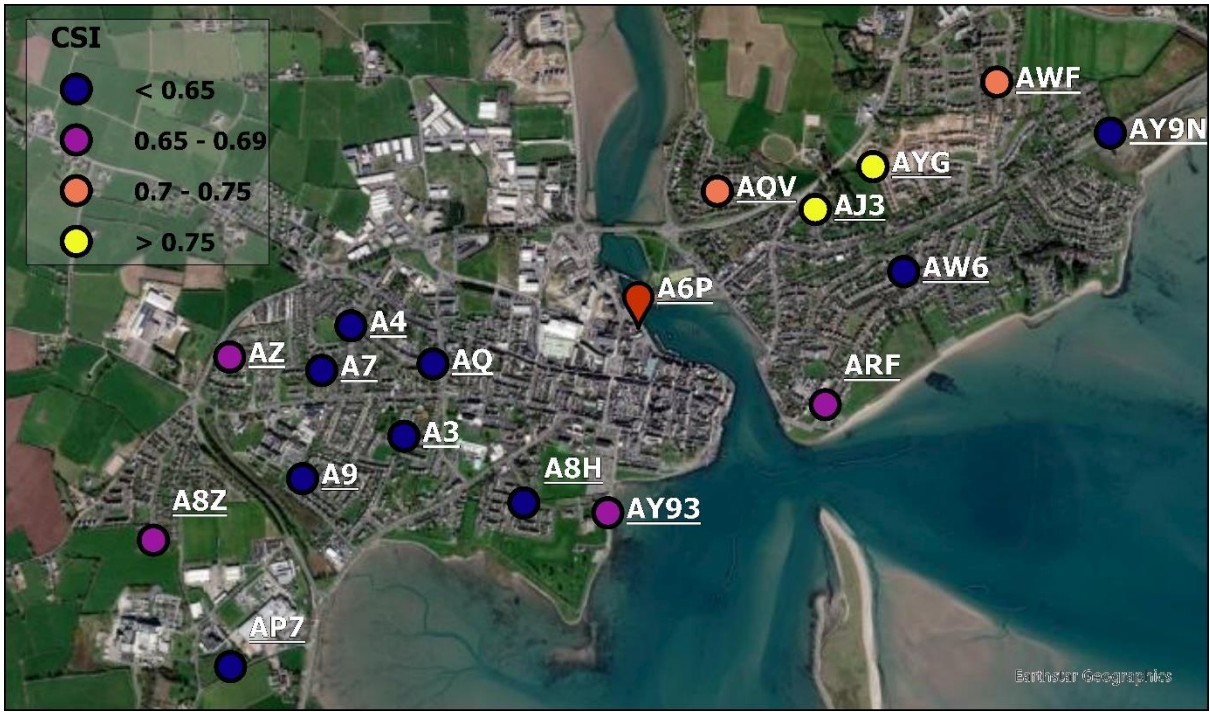

**Figure 9: Dungarvan AQS locations with CSI results indicated in coloured circles (blue = lowest CSI, yellow = highest CSI), and**
**A6P location indicated with red pin marker. (Map obtained from Esri, DigitalGlobe, GeoEye, i-cubed, USDA FSA, USGS, AEX, Getmapping, Aerogrid, IGN, IGP, swisstopo, and the GIS User Community)**

One of the devices in the Cork City senor network, UCC, was co-located alongside a reference instrument (BAM-1020) at

the national air quality monitoring location on UCC campus. The CSI values for device labelled UCC are shown in Fig. 10,

showing how similar the measurements at this site are compared to the rest of the locations in the sensor network. The CSI

scale on the map has been adjusted for these values. Similar to the Dungarvan case, there are devices which show among the

highest similarity (CCC4, CCC2, CCC12, CCC1) with UCC which are not located nearby.





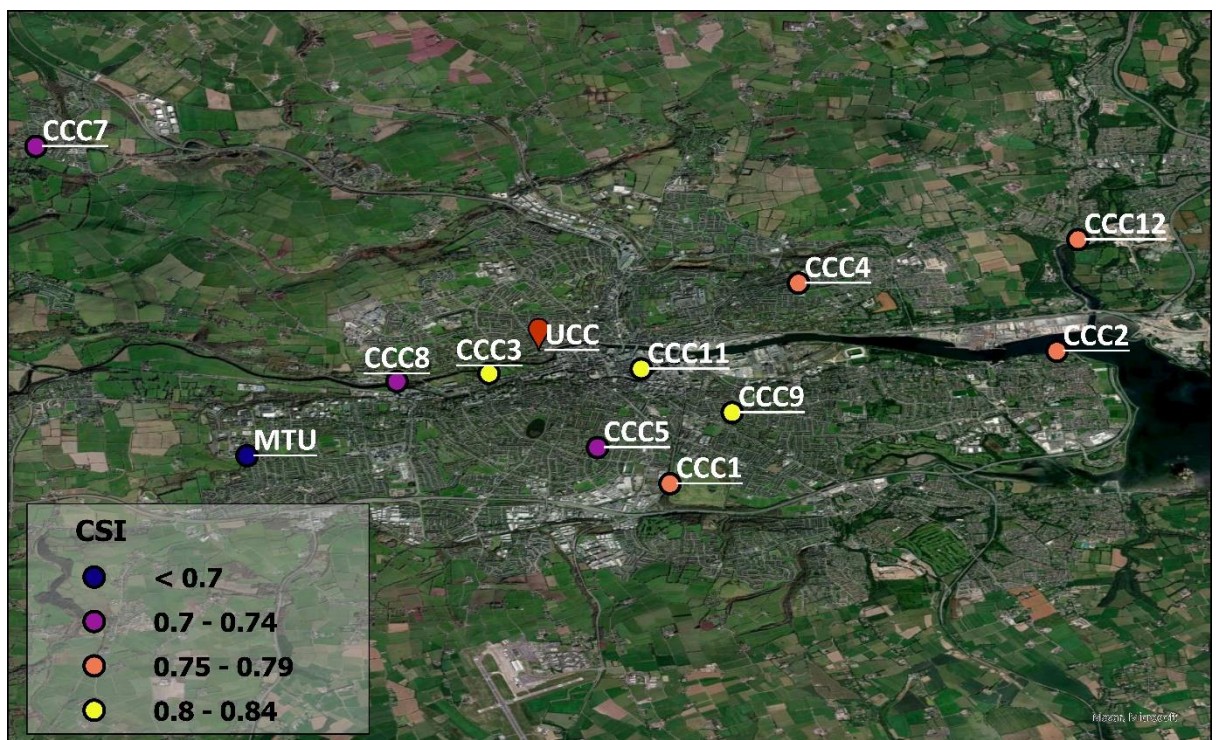

**Figure 10: Cork AQS locations with UCC CSI results indicated in coloured circles (blue = lowest CSI, yellow = highest CSI), and UCC location indicated with red pin marker. (Map obtained from Esri, DigitalGlobe, GeoEye, i-cubed, USDA FSA, USGS, AEX, Getmapping, Aerogrid, IGN, IGP, swisstopo, and the GIS User Community)**

**4 Conclusion**

A robust framework for comparing data series from individual air quality sensors in a network has been established and a new metric, the Concentration Similarity Index (CSI), has been developed, optimised, and tested on a co-location dataset. The CSI allows one to consider the monitoring network in terms of the similarity of the concentration-time profile of $PM_{2.5}$ at one location to those at the other locations in the same network. The harmonised dataset with minimal unexplained inter-sensor variation underpins the development of the CSI method, along with robust tests to ensure that the function represents an unbiased and fair depiction of the inter-sensor relationships after deployment in a monitoring network.

The CSI method has been used to analyse data generated by $PM_{2.5}$ sensor networks in two locations in Ireland, the coastal town of Dungarvan and the city of Cork. Clustering techniques are applied to the CSI matrix and comparable similarity trends between locations drives the distinctions made with the clustering algorithm. The resulting groupings can provide several insights into the $PM_{2.5}$ profile at each location, including the likelihood of similarity in pollution sources, spatial patterns, and temporal trends. An interesting contrast in the CSI results from the two monitoring networks was obtained from the clustering analysis. In Dungarvan, the locations generated clusters that were well reflected when comparing the individual diurnal profiles and specifically the diurnal maximum values, indicating that this factor has a major influence



when relating the concentration-time profiles at each location to one another in this network. However, for the Cork City network results, this was not as apparent. The clusters were not aligned based on diurnal peaks but rather the differentiating factor was more nuanced. Both clusters contained locations with a mix of higher and lower diurnal maxima and overall maxima. However, both network groupings reflect that devices may report dissimilar CSI results to other devices located nearby, and that considering location specifications or types, such as residential areas, is more important than physical

proximity when it comes to understanding and quantifying the similarities between locations.

The CSI function was also applied to two separate months in the network datasets, with January chosen to represent a period of higher $PM_{2.5}$ levels due to solid fuel burning emissions, and May chosen to represent a period with lower $PM_{2.5}$ concentrations due to reduced solid fuel burning. In both locations, the mean CSI for the network comparisons was higher in May than in January, indicating that higher $PM_{2.5}$ levels is a major driver for lower similarity indices between sensor

locations. Combining this with the findings of our previous study, we provide further evidence that high levels of localised $PM_{2.5}$ cause distinct disparities in exposure to poor air quality in different locations. Furthermore, to properly assess the burden of $PM_{2.5}$ experienced by a population and to accurately compare the measurements at two locations, the wintertime PM data must be included in the assessment.

The similarity of $PM_{2.5}$ measured at designated sites in the national air quality monitoring network compared to the rest of

locations in the sensor networks was analysed to give an estimation of the representativeness of the air pollution measured at the designated monitoring site. The national monitoring site location in Dungarvan was shown to be moderately representative of the other AQS network locations in the town, with CSI values ranging from 0.53 to 0.72. The CSI values for the Cork City comparison ranged from 0.62 to 0.84, also showing a fair representation of the air pollution experienced in the rest of the network. The CSI function was also tested via synthetic datasets which showed that a positive offset of just 5

µg m$^{-3}$ resulted in almost halving the CSI, which was a lower CSI than most of the sensor comparisons in both network locations. So, while a CSI of 0.85 was used as a limit for two sensor measurement sets being very similar, CSI values between 0.6 and 0.7 are still moderately similar. In general, the CSI values in Cork City for the reference site comparison were higher (mean = 0.75) than that of Dungarvan (mean = 0.63), indicating less similarity between the reference site and devices in the Dungarvan network compared to Cork City. This analysis and application of the CSI function displays the

potential for AQS networks to be used in conjunction with a regulatory monitoring system. There is potential for the application of sensor networks to assess the need for more regulatory monitoring in an area, and to identify locations that are being poorly represented by the current system. Furthermore, the CSI method can be used to optimise a sensor network by carrying out short term sensor deployments and identifying areas of similarity or dissimilarity and thus assessing where the best locations for sensors are based on the similarity in exposure to air pollution.






**Data Availability**

All raw data are available upon request.

**Author Contributions**

RB and SH conceptualised the project and the methodology. RB carried out the formal data analysis, investigation, visualisation, and wrote the manuscript with supervision and contributions from SH and JCW.

**Competing Interests**

The authors declare that they have no conflict of interest.

**Acknowledgements**

The research carried out was supported by the EPA and DECC and the EU LIFE Programme through LIFE Emerald – LIFE19 GIE/IE/001101. The authors also acknowledge Cork City Council, especially Kevin Ryan, for developing and maintaining the air quality sensor network in Cork City, and Waterford City & County Council for supporting the Dungarvan measurement campaign. In particular, Paul Flynn who facilitated the physical deployment of the air quality units in Dungarvan.



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
