# Peer review of "Spatial analysis of PM2.5 using a Concentration Similarity Index applied to air quality sensor networks"

_Atmospheric Measurement Techniques, 2024_

## Author Response (AR1)

**RC1**

**Thank you for the review and very helpful comments. Please find our replies below. Note that line numbers in our replies refer to the revised manuscript.**

> **Comment**: Review of Spatial analysis of PM 2.5 using a Concentration Similarity Index applied to air quality sensor networks by Byrne, Wenger and Hellebust
>
> This paper introduces the Concentration Similarity Index (CSI) and uses it to evaluate air quality data from networks of low cost PM sensors in Dungarvan and Cork Ireland. The CSI yields a quantitative and time-averaged method of comparing measurements from each site within the network. The authors have done a commendable job by calibrating and deploying these networks and analysing the data. I would like to see additional discussions and conclusions regarding the CSI method, how it compares to other techniques, advantages/disadvantages and when and why the authors would recommend its use. In addition the manuscript could be improved by rewriting to underpin descriptive/qualitative statements with numbers, and with a careful proofreading to somewhat reduce the word count and improve clarity. Overall it is a very nice study and I recommend publication after minor revision to address these matters and the points below.

**Reply:** Thank you for your comment. We have added quantitative information to underpin descriptive/quantitative statements (Line 15, 20, 23, 24, 30, 31, 143-146, 308, 360, 381, 382, 434, 450, 451, 473). We have also added further discussion on the robustness of the CSI compared to correlation assessments (Line 272 – 285, Table 3). We removed some sections of text to reduce the overall word count, such as the more in-depth explanation of hierarchical and fuzzy c-means clustering.

> **Comment**: It would be useful to include more meteorological data in the analysis, such as a wind rose for Dungarvan and Cork, and statistics regarding wind direction and speed. I would imagine this information is available from public agencies and then speculative statements can be avoided e.g. Line 16, 'possibly due to the town's coastal location'. I could guess that this is because it might be windier at the coast but the reader shouldn't have to guess. If you mean wind speed say it and even better, provide some data.

**Reply**: Thank you for the suggestion. A brief discussion of meteorological data has been added in both the Dungarvan (lines 327 – 336, Figures S4 and S5) and Cork (lines 411-419, Figure S8) results sections and subsequently additional information has been provided in the methodology section (Pages 5-6, lines 154 – 164).

> **Comment**: I would like more discussion evaluating the new index the CSI. How does it compare to other approaches like deviation between a given sensor and network

average behavior, or comparison of a low cost sensor with a monitoring station, or Fourier Transform and filtering? What are the advantages and disadvantages? Is CSI well suited to networks of low cost sensors with their specific behaviours and calibration issues? Would you recommend widespread adaptation of the CSI, and on what basis?

**Reply**: While we did include a statistical evaluation of both network datasets in the original manuscript and discussed the suitability of the CSI to sensor networks, we have now extended the discussion regarding the comparison of the CSI to the correlation between two sensors. Specifically, we discuss potential limitations of the CSI (lines 535 – 538) and how the correlation is not as robust to outliers and does not give a realistic reflection if there is a bias or offset present between a pair of measurements as these can still have a high correlation (line 272 - 285). We have also added additional synthetic data tests regarding the addition of noise to the data and a comparison of the CSI and $R^2$ response (Table 3).

**Comment**: At times the manuscript says things qualitatively which could instead be said quantitatively. By presenting the evidence you allow the reader to form their own judgement. One example is that I would like to see numbers in the abstract - specific concentrations and CSI values. Also, line 138 'low inter-sensor and inter-unit variability was exhibited by four ..devices.' -- the reader is left wondering what 'low' means, and how do the inter-sensor and -unit variabilities compare. Better to define low and present the numbers. The statement at line 302, 'Although the PM 2.5concentrations are not as accurate as those collected by reference instrumentation, any relative differences between the sensors and individual sensor data trends can be regarded as genuine due to the low inter-sensor variation observed after data harmonisation procedures.' could be made quantitative by providing values.

**Reply:** As stated above, we have now added quantitative information to underpin descriptive/quantitative statements (Line 15, 20, 23, 24, 30, 31, 143-146, 308, 360, 381, 382, 434, 450, 451, 473) to improve clarity for the reader.

**Comment**: I would like to see some specific data-driven conclusions regarding the air quality in Dungarvan and Cork, rather than:

Line 19 'locations in central or residential areas which experience more pollution from sold fuel burning and locations on the edge of the urban areas which experience cleaner air.' Please present data -- how much more?
Line 26 'The findings of this work underscore the influence of solid fuel combustion as a local contributor to PM 2.5 and the variation it can cause between the measurements at different monitoring locations in a network while also highlighting the importance of including wintertime PM data for accurate comparisons.' What are the numbers, how important of a local contributor, how much variation?
Line 29 'The CSI method developed here could be a valuable tool for quantitative comparisons of air quality within a monitoring network, offering insights for further regulatory monitoring and exposure assessments.' Please demonstrate this - give

examples of these comparisons, in order to show that it is a valuable tool. What are these insights? Better to present them than to say that they could exist.

**Reply:** In this paper we have not developed a source apportionment model, and as such we do not have estimates of the contribution of solid fuel combustion to local air pollution. However, we can cite the measured variation between locations and so we have included values for the differences in mean $PM_{2.5}$ measurements derived from clustering (line 20) and the differences in CSI averages for winter and summer periods (line 23, 24) to highlight the seasonal effect. We have clarified the statements to make it clear that we are emphasising the importance of including wintertime PM data as the variations between locations is more enhanced when solid fuel burning activities are likely at their peak, not because we are estimating local source contributions. Quantitative information is also added (lines 30, 31) to reflect our findings regarding the assessment of the regulatory monitoring location.

**Comment**: Recommend adding a discreet legend indicating 'North' to the maps.

**Reply:** Added to maps (Figures 4, 8, 9, 10).

**Comment**: Line 88, 'what is the extent of the geographical area that the location meaningfully represents' - I think there is a term for this, the footprint, that could be used to increase clarity/reduce length.

**Reply:** Thanks. On reflection, and in keeping with efforts to reduce the text, this statement has been removed from the revised manuscript (lines 91- 93).

**Comment**: Line 96, 'The highest sampling frequency..was 8 minutes' but consider that a frequency has dimensions of inverse time; the sampling _interval_ is 8 minutes.

**Reply:** The text has been corrected as suggested.

**RC2**

**Thank you for your helpful review. Please find our replies below. Note that line numbers in our replies refer to the revised manuscript.**

**General comments:**

the paper addresses an important aspect of air quality monitoring, i.e the limited spatial representativeness of monitoring sites, even with low cost air quality sensors which in principle allow to increase the number of monitoring points with respect to reference instruments.

The paper is well structured and written in very clear language. It is very long and didactic, which makes the reading quite long, so I suggest to shorten and condense some sections here and there, avoiding to repeat similar contents in consecutive sentences.

**Reply:** Thank you for your thorough review. We have taken your suggestion and condensed certain sections to avoid repetition and reduce the overall wordcount, such as the removal of the more in-depth explanation of hierarchical and fuzzy c-means clustering.

**Comment**: The main limit is the generality of the proposed index. Having this been developed on 1 network of 18 co-located sensors in Dungarvan, and validated on another network of 4 co-located sensors in Cork, I believe that the index, with the provided parametrizations of $PM_{lim}$, $C_{lim,upper}$ and $C_{lim,lower}$ , is quite case-sensitive. This is not bad per se, but I expect that the authors clearly state this limitation in the discussion, if they do not have further data to validate the index.

**Reply:** This is a good point. A sentence has been added to the conclusion (lines 535 – 538) to recommend validation of the parameters on a co-location dataset before wider application of the CSI to a network of sensors. This co-location step is also now recommended to ensure minimal inter-sensor variability.

Specific comments:

lines 15-17: it is not clear why the coastal location should give rise to higher variation in the network and this seem not to be recalled in the text

**Reply:** Since we cannot substantiate this statement, the text "possibly due to the town's coastal location giving rise to higher variation within the network" has been removed from the abstract.

**Comment**: Line 20: sold à solid

**Reply:** Thank you, this has been fixed.

**Comment**: Line 72: the short-term (hourly-dalily) impact on health should be mentioned and adequately referenced here. It is correct that an annual average is an incomplete representation of true exposure, if the hourly and dailiy variability is high, but 1) I would not define it as "poor" (yearly average and horly-daily values are correlated) and and most of all 2) the reason is the short-term impacts

**Reply:** We have amended the text (lines 71-78) to reflect that the annual average is not a "poor" representation, but rather an incomplete one. Additional references also added.

**Comment**: Line 89: a comma between to and regardless would help

**Reply:** The text has been altered as suggested.

**Comment**: Line 90: delete the question mark

**Reply:** The text has been altered as suggested.

**Comment**: Lines 154-166: Piersanti et al. (2015) did not use point measurements. They only used modelled values, with the point measurement location used as reference. Even if this does not modify the general meaning of the text block, I recommend to correct.

**Reply:** Thank you. The text has been clarified (lines 173 – 175).

**Comment**: Lines 185-186: you need to explain why, i.e. the network in Piersanti et al. (2015) comes from a regional model covering some hundreds of kilometers, your network covers few kilometers, and the pollution dynamics are completely different.

**Reply:** Changes have been made here to improve clarity, in particular on page 8, lines 200 - 204.

**Comment**: Lines 187-189: again, you need to be more precise: the original function has a part (the mathematics) independent from the local conditions, and another (i.e.the threshold values) which is arbitrary and can be 5dapted.

**Reply:** We have made changes to improve clarity of the text (lines 204-207).

> **Comment**: Lines 217-219: here you have cited A and B without defining them before. You say that they are in Eq. 1, which is not true. So, either you rearrange the text defining explicitly A and B, or you move this section after the Eq. 2.

**Reply:** The text has been altered as suggested. $C(A, t_i)$ and $C(B, t_i)$ were referred to in line 170 (as per the original manuscript) as being substitutions for the concentration variables in Eq. 1. but A and B were not explicitly defined. This has been corrected (lines 188 - 190), and it has also been clarified on line 222.

> **Comment**: Eq. 2: the 4 cases have partial overlappings, I believe there are 2 typos (the $<$ sign in case 2 and the $>$ sign in case 3). Furthermore, you should leave $C_{lim,upper}$ and $C_{lim,lower}$ explicitly indicated, rather than writing the 0.2 and 0.7 values.

**Reply:** Thank you. This has been fixed and we have left the threshold values in the equation explicitly indicated.

> **Comment**: Line 231: 0.2 and 0.7 are very different, this affecting the treatment of concentration values around $PM_{lim}$ , changing depending on being lower or higher. Even if this would have needed some quantitatve assessment, I recommend at least to cite this limitation, somewhere in the text.

**Reply:** The iterative selection process for $PM_{lim}$, $C_{lim,upper}$, and $C_{lim,lower}$ is described on page 9 with the reasoning for the addition of $PM_{lim}$ also discussed on page 8 (Section 2.2.2). However, we have added the word "co-located" to line 244 to improve the clarity of the description of the selection process for the different threshold values.

> **Comment**: Line 245: why did you not repeat the A6P test on synthetic data on all sensors? If just for matter of time/resources, please add it somewhere as a limitation of the test.

**Reply:** We did carry out the same tests for all devices in the 18-sensor co-location assessment. However, the differences were small and we did not deem it necessary to include the values for all 18 comparisons. We have now added the following text (lines 268 – 271) stating that these differences were small and the same trends were seen for each sensor across all synthetic data comparisons: "Low variations were found during all synthetic data analyses with the resulting CSI values having standard deviations $\leq 0.05$ across the individual devices for each test. As an example, the effects of these tests on the CSI results for A6P are shown in Table 3, where 4406 data points were included in the calculations."